# Age-associated mortality is partially mediated by *TERT* promoter mutation status in differentiated thyroid carcinoma

**Jung Heo[1‡], Sungjoo Lee[2‡], Jun Park[3], Heera Yang[4], Hyunju Park[5], Chang-Seok Ki[6], Young Lyun Oh[7], Hye In Kim[8], Sun Wook Kim[4], Jae Hoon Chung[4], Kyunga Kim[2,9,10‡]\*, Tae Hyuk Kim[4‡]\***

1 Division of Endocrinology and Metabolism, Department of Internal Medicine, Myongji Hospital, Hanyang University College of Medicine, Goyang-si, Gyeonggi-do, Korea, 2 Department of Digital Health, Samsung Advanced Institute for Health Sciences and Technology, Sungkyunkwan University, Seoul, Korea, 3 Division of Endocrinology, Department of Medicine, Sahmyook Medical Center, Seoul, Korea, 4 Division of Endocrinology and Metabolism, Department of Medicine, Samsung Medical Center, Sungkyunkwan University School of Medicine, Seoul, Korea, 5 Department of Internal Medicine, CHA Bundang Medical Center, CHA University, Seongnam-si, Gyeonggi-do, Korea, 6 Green Cross Genome, Yongin-si, Gyeonggi-do, Korea, 7 Department of Pathology, Samsung Medical Center, Sungkyunkwan University School of Medicine, Seoul, Korea, 8 Division of Endocrinology and Metabolism, Department of Medicine, Samsung Changwon Hospital, Sungkyunkwan University School of Medicine, Changwon-si, Gyeongsangnam-do, Korea, 9 Biomedical Statistics Center, Research Institute for Future Medicine, Samsung Medical Center, Seoul, Korea, 10 Department of Data Convergence and Future Medicine, Sungkyunkwan University School of Medicine, Seoul, Korea

‡ JH and SL contributed equally to this work as co-first authors. THK and KK also contributed equally to this work as co-corresponding authors.
\* taehyukmd.kim@samsung.com (THK); kyunga.j.kim@samsung.com (KK)

**Data Availability Statement:** Data cannot be publicly shared because it contain sensitive or potentially identifying patient information. Data are

## Abstract

### Background

Age at diagnosis (AAD) and telomerase reverse transcriptase (*TERT*) promoter mutations are prognostic factors in differentiated thyroid carcinoma (DTC), and the prevalence of the mutations increases with AAD. Considering this correlation, we investigated whether an interaction between AAD and the mutations is present and whether the mutation mediates the effect of AAD on the mortality rate in DTC.

### Methods

The study included 393 patients with DTC who were followed-up after thyroidectomy at a single medical center in Korea from 1994 to 2004. Multivariable Cox regression was used to investigate the interaction of AAD and *TERT* promoter mutation. Mediation analysis was conducted using a regression-based causal mediation model.

### Results

The age-associated mortality rate increased progressively in all DTC patients and wild-type *TERT* group (WT-*TERT*) with a linear trend ($p < 0.001$) contrary to mutant *TERT* group (M-*TERT*) ($p = 0.301$). Kaplan-Meier curves declined progressively with increasing AAD in the

available upon reasonable request from the Samsung Medical Center Institutional Data Access/Ethics Committee (The Institutional Review Board of Samsung Medical Center, http://www.samsunghospital.com/dept/main/index.do?DP_CODE=CEC) for researchers who meet the criteria for access to confidential data.

**Funding:** The author(s) received no specific funding for this work.

**Competing interests:** The authors have declared that no competing interests exist.

entire group, but the change was without significance in M-*TERT*. The effect of AAD on mortality was not significant (adjusted HR: 1.07, 95% CI 0.38–3.05) in M-*TERT*. An interaction between AAD and *TERT* promoter mutation ($p = 0.005$) was found in a multivariable Cox regression. *TERT* promoter mutations mediated the effect of AAD on the mortality rate by 36% in DTC in a mediation analysis.

## Conclusions

Considering the mediation of *TERT* promoter mutation on the effect of AAD on mortality, inclusion of *TERT* promoter mutation in a stage classification to achieve further individualized prediction in DTC is necessary.

## Introduction

Age at diagnosis (AAD) is a well-known critical prognostic factor in differentiated thyroid carcinoma (DTC) [1–3]. Age-associated cancer-specific mortality in thyroid cancer is incorporated in 8th edition of the American Joint Committee on Cancer (AJCC) with a cutoff of 55 years implying AAD ≥55 years presents a worse cancer-specific mortality than AAD <55 [4]. Poor outcome in DTC patients with older AAD has been explained with several factors such as decreased capacity of immune surveillance, increased TSH level, decline in radioiodine ablation (RAI) responsiveness, and high prevalence of *BRAF* V600E and telomerase reverse transcriptase (*TERT)* promoter mutation [5–9]. Despite these comprehensive theories, to what extent each factor contributes to mortality in DTC is yet remained to be elucidated.

*TERT* promoter mutations are related to cancer aggressiveness and act as an independent poor prognostic factor in DTC [10–14], and the prevalence of *TERT* promoter mutation is higher in older AAD [15, 16]. In recent studies, new staging/risk stratification systems or classifications of DTC incorporating *TERT* promoter mutation have shown improved predictability of recurrence or cancer-specific survival (CSS) [17–19], which emphasized the prognostic value of *TERT* promoter mutations in DTC.

Given that AAD and *TERT* promoter mutations are both independent prognostic factors for cancer-specific mortality, and the likelihood of high prevalence of *TERT* promoter mutation in older AAD, our study group sought the correlation of AAD and *TERT* promoter mutation in cancer-specific mortality. Specifically, we investigated whether an interaction between AAD and *TERT* promoter mutation was present, along with the manner and extent of the interaction. We also hypothesized that *TERT* promoter mutation would mediate the effect of AAD on cancer-specific mortality in DTC and elicited the extent of mediation by *TERT* promoter mutations using mediation analysis, which is used to discover variables besides exposure and outcome and to investigate the detailed determinants of cancer-specific mortality [20–22].

## Materials and methods

### Patients and clinicopathological data

The dataset from the previous studies of our study group was used in this study [18, 23]. In short, we included 393 patients with pathologically confirmed DTC from October 1994 to December 2004, 327 papillary thyroid cancer (PTC) patients, and 66 follicular thyroid cancer (FTC) (including Hurthle cell carcinoma) patients after thyroidectomy and neck dissection.

All patients received thyrotropin suppression therapy, and 364 patients received RAI ablation postoperatively according to standard guidelines [24, 25].

Because 'age at diagnosis' is frequently used in this article, we represented the term using the abbreviation 'AAD', as used in previous study [26].

Mutation analyses were conducted at the Department of Pathology of Samsung Medical Center, with one sample taken from the thyroid cancer tissue of each patient. The results of the mutation analyses did not affect the decision-making process of the physicians because this analysis was performed after surgery and RAI treatment. Thyroid cancer-specific mortality data were acquired from the Korea National Statistical Office and hospital medical records.

### Detection of *TERT* promoter mutations

Genomic DNA was extracted from formalin-fixed, paraffin-embedded (FFPE) tissue using a Qiagen DNA FFPE Tissue Kit (Qiagen, Germany) according to the manufacturer's instructions. 4-μm-thick unstained slides from the FFPE tissue were prepared, and the slides with a minimum 75% tumor component were selected for DNA extraction. Then, we performed a semi-nested polymerase chain reaction (PCR) to identify *TERT* promoter mutations and mutant enrichment with 3'-modified oligonucleotide-PCR.

### Statistical analysis

The mortality rate in DTC patients was compared between wild-type *TERT* (WT-*TERT*) and mutant *TERT* (M-*TERT*) groups as AAD (i.e., $\leq 45$, 45–55, 55–65, $\geq 65$) increased to identify any association between AAD and *TERT* promoter mutation status, and Cochran-Mantel-Haenszel (CMH) test was used to compare the mortality rate between WT-*TERT* and M-*TERT* groups. Cochran-Armitage trend test was used to examine the linear trend of mortality rate with increasing AAD.

The period of thyroid CSS for survival analysis was defined as the time from the initial surgical treatment to the date of thyroid cancer-specific death or the last observation date (December 31, 2018). Patients who died from other causes were considered censored at the time of death. The CSS by AAD group was analyzed by Kaplan-Meier survival curves with log-rank tests. It was performed for all DTC patients, the WT-*TERT* group, and the M-*TERT* group.

Univariable and multivariable Cox proportional hazards models were used to estimate the unadjusted and adjusted hazard ratios (HRs) and 95% confidence intervals (CIs). The covariates in multivariable analysis were variables with $p$-value $\leq 0.2$ in univariable analysis, and the selected covariates were sex, AAD, *TERT* promoter mutation, histological type, extrathyroidal extension (ETE), distant metastasis, tumor size, and total dose of RAI. Firth's penalized option was used in all multivariable Cox regression analyses to correct any unbalanced patient distribution of covariates.

To identify the role of *TERT* promoter mutation as a modifier of the effect of AAD in thyroid CSS, we included an interaction term (i.e., AGE×*TERT* promoter mutation) in the multivariable Cox proportional hazards model to investigate whether there was evidence of a significant interaction. In addition, subgroup analyses by *TERT* promoter mutation were conducted using a Cox proportional hazards model to compare the magnitude of the AAD effect for thyroid CSS between subgroups. Smooth HR curves were displayed to demonstrate the continuous relationship between AAD and cancer-specific mortality. Additive Cox models with penalized spline method were applied to provide HR curves and 95% confidence bands taking the reference value as the AAD of 55 [27]. All statistical analyses were performed using R 3.6.1 (Vienna, Austria; http://www.R-project.org). A $p$ value $\leq 0.05$ was considered statistically significant.

## Mediation analysis

Regression-based causal mediation model with a treatment-mediator interaction proposed by VanderWeele was used in our mediation analysis [28–32]. We set the exposure, mediator, and outcome as AAD, *TERT* promoter mutation, and mortality, respectively. The multivariable logistic and Cox regression models were used to evaluate the association between exposure and mediator and the associations of outcome with exposure and with mediator, respectively. Sex, ETE, histologic type, tumor size, and RAI total dose were included as confounders which showed a *p*-value ≤ 0.2 in univariable analyses, and there were no missing data.

Our hypothesized model of mediation is schematized in Fig 1. In this three-way decomposition, we calculated the pure direct effect (PDE) (Pathway 1), the pure indirect effect (PIE) (pathway 2 to 4), and the total effect (TE) (the sum of Pathways 1, 2 to 4, and 2 to 3). PDE indicates the direct effect of AAD on mortality rate without mediation. PIE indicates the effect of AAD on mortality rate solely through mediation by *TERT* promoter mutation. In addition, when AAD effects mortality rate through *TERT* promoter mutation, the degree of the effect is altered due to moderation of *TERT* promoter mutation. Mediated interaction (INT_med) (Pathway 3) denotes this alteration of the AAD effect on mortality rate by *TERT* promoter mutation as a mediator. The total indirect effect (TIE) demonstrates the sum of PIE and INT_med. The total effect (TE) implies the sum of PDE and TIE. The proportion_mediated was calculated using PDE and TIE. The formula to calculate the proportion_mediated is PDE (TIE—1)/(PDE x TIE -1) [31].

To explore the potential effects of unmeasured confounders, we conducted a sensitivity analysis with E-values for the pure indirect effect and its lower limit of the 95% CI [32]. R package was used for mediation analysis. All 95% CIs were derived via bootstrapping with 1000 replicates.

## Results

### Baseline characteristics

The clinicopathological characteristics of patients with DTC are summarized in Table 1. Of the total 393 patients, 329 (83.7%) were female, and 64 (16.3%) were male. The median AAD

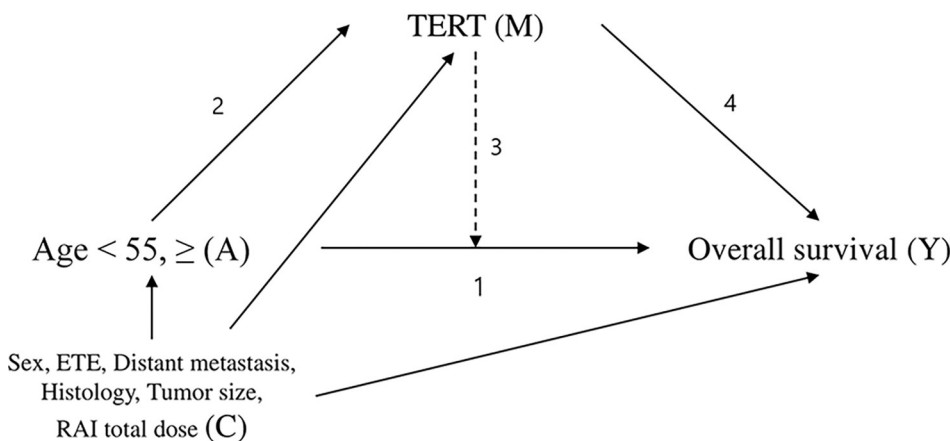

**Fig 1. A diagram of the mediational pathway in this study.** A (exposure) is age at diagnosis ≥ 55, M (mediator) is *TERT* promoter mutation, Y (outcome) is overall survival, and C (confounders) represents multiple confounders as sex, extrathyroidal extension (ETE), distant metastasis, histology, tumor size, and radioactive iodine (RAI) total dose. Pathway 1, the arrow from A to Y, represents the pure direct effect (PDE). Pathways 2 to 4, the arrows from A to Y via M, represent the pure indirect effect (PIE). Pathway 2 to 3, arrow A to Pathway 1 via M, indicates the mediated interaction (INT_med).

**Table 1. Baseline characteristics of patients with differentiated thyroid cancer (DTC).**

| Characteristic | N (%) |
|---|---|
| Sex | |
| Female | 329 (83.7) |
| Male | 64 (16.3) |
| Age at diagnosis, y | |
| Median (range) | 42.8 (15.8–81.4) |
| Tumor size, cm | |
| Median (range) | 2.7 (0.4–12.0) |
| Histological type | |
| PTC | 327 (83.2) |
| FTC | 66 (16.8) |
| Multifocality | |
| Absent | 286 (72.8) |
| Present | 107 (27.2) |
| Lymph node metastasis | |
| Absent | 199 (50.6) |
| Present | 193 (49.1) |
| Missing data | 1 (0.3) |
| Extrathyroidal extension | |
| Absent | 352 (89.6) |
| Present | 41 (10.4) |
| Distant metastasis | |
| Absent | 370 (94.1) |
| Present | 23 (5.9) |
| AJCC TNM stage at diagnosis | |
| I | 329 (83.7) |
| II | 48 (12.2) |
| III/IV | 16 (4.1) |
| *TERT* promoter mutations | |
| WT | 350 (89.1) |
| Mutant | 43 (10.9) |
| C228T | 39 (9.9) |
| C250T | 4 (1.0) |
| Radioactive iodine treatment | |
| Absent | 29 (7.4) |
| Present | 364 (92.6) |
| Death | |
| Survival | 366 (93.1) |
| Death | 27 (6.9) |

Abbreviations: *TERT*, telomerase reverse transcriptase; PTC, papillary thyroid cancer; FTC, follicular thyroid cancer; AJCC, American Joint Committee on Cancer.

was 42.8 years (range: 15.8–81.4 years); 327 (83.2%) were patients with PTC, and 66 (16.8%) were patients with FTC. Of all patients, 286 (72.8%) had unifocality, and 329 (83.7%) were stage I at diagnosis. Finally, 92.6% of all patients received postoperative radioactive iodine therapy, and *TERT* promoter mutation was found in 43 (10.9%) patients. During the follow-up period of 16 years (interquartile range: 14–19 years), 27 (6.9%) cancer-related deaths were identified.

### Age-associated mortality rate patterns in the WT-TERT and M-TERT groups

As shown in Fig 2, the mortality rates were low in the WT-*TERT* compared with the high mortality rate in the M-*TERT* group. In Cochran-Mantel-Haenszel (CMH) test, the mortality rate of M-*TERT* was significantly higher than WT-*TERT* group ($p = 0.005$). A linear association was found between patient AAD and mortality rate in all patients with DTC and in the WT-*TERT* group as AAD increased ($p < 0.001$ for trend). The association did not have a linear pattern in the M-*TERT* group as seen in the WT-*TERT* group, although the mortality rate increased generally with AAD ($p = 0.301$ for trend).

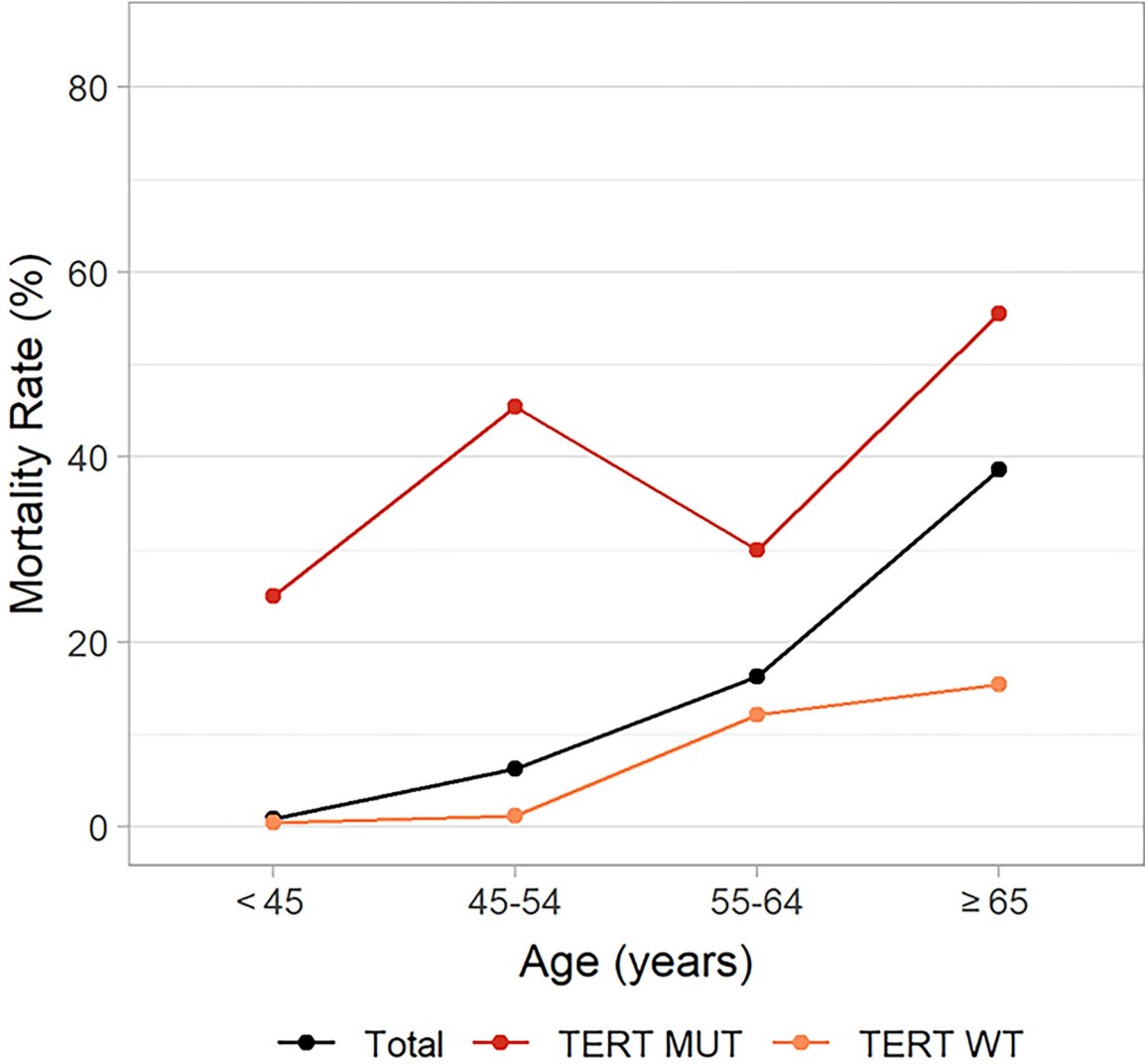

**Fig 2. The relationship between age at diagnosis and differentiated thyroid cancer (DTC)-specific mortality rate in all patients with DTC and in the wild-type *TERT* (WT-*TERT*) and mutant *TERT* (M-*TERT*) groups.** Mortality rates according to age at diagnosis (AAD) are shown for three groups; total patients, wild-type *TERT* (WT-*TERT*), and mutant *TERT* (M-*TERT*). Yellow curve represents mortality rate of WT-*TERT*, and black curve represents that of total patients with differentiated thyroid cancer (DTC). Mortality rates of WT-*TERT* and total patients with DTC increased with AAD ($p < 0.001$ for trend). Red curve represents mortality rate of M-*TERT*, and it is higher compared to WT-*TERT* at any AAD. However, mortality rate of M-*TERT* did not show any linear association with AAD ($p = 0.301$ for trend).

## Comparison of Kaplan-Meier survival curves according to TERT mutation status

The Kaplan-Meier survival curves declined progressively with increasing AAD in all patients with DTC (log-rank $p < 0.001$) and in the WT-*TERT* group (log-rank $p < 0.001$) (Fig 3A–3F). Although the Kaplan-Meier survival curve of the M-*TERT* group declined with increasing AAD, statistical significance was not obtained in the log-rank test ($p = 0.500$ in an analysis with four AAD groups; $p = 0.927$ in an analysis with two AAD groups) (Fig 3B and 3E).

## Interaction of age at diagnosis and TERT promoter mutation

We conducted univariable Cox analyses in DTC and PTC subgroups, and a multivariable Cox regression analysis in patients with DTC of the 10-year survival rate. In univariable analyses, lymph node metastasis ($p = 0.964$ in DTC, $p = 0.410$ in PTC) and *BRAF* V600E mutation ($p = 0.331$ in DTC, $p = 0.253$ in PTC) were not selected to multivariable analysis with $p$-

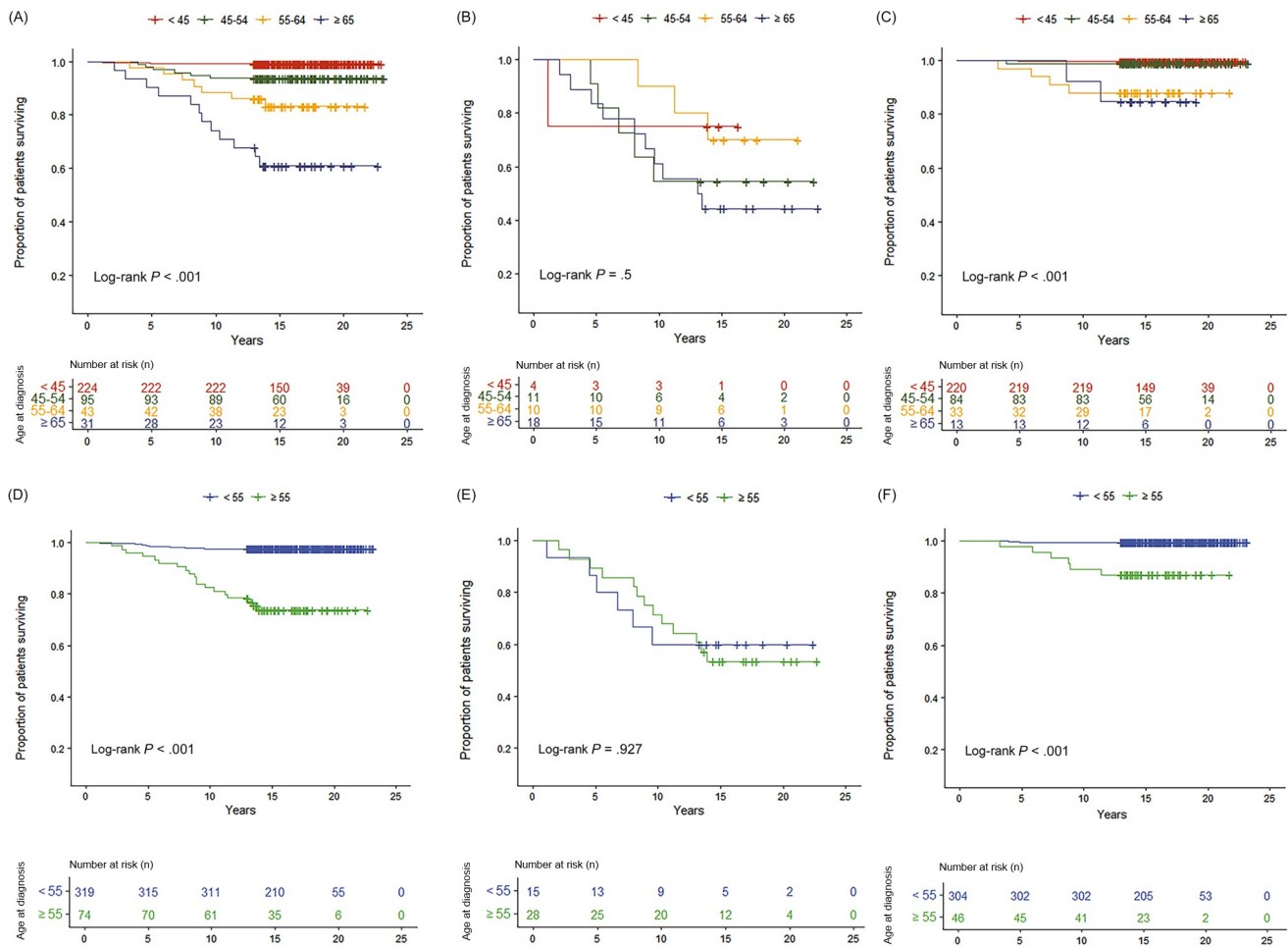

**Fig 3. The Kaplan-Meier survival curves of cancer-specific survival (CSS) in patients with differentiated thyroid cancer (DTC) in various age at diagnosis groups.** Each Kaplan-Meier survival curves represents cancer-specific survival of different age at diagnosis (AAD) groups in total patients with differentiated thyroid cancer (DTC), mutant *TERT* (M-*TERT*), and wild-type *TERT* (WT-*TERT*). (A) Kaplan-Meier survival curves of total patients with DTC, (B) Kaplan-Meier survival curves of M-*TERT*, and (C) Kaplan-Meier survival curves of WT-*TERT* stratified into four AAD groups: <45, 45–54, 55–64, ≥65. (D) Kaplan-Meier survival curves of total patients with DTC, (E) Kaplan-Meier survival curves of M-*TERT*, and (F) Kaplan-Meier survival curves of WT-*TERT* stratified into two AAD groups: <55 and ≥55.

values > 0.2 (S1 and S2 Tables). In multivariable analysis of patients with DTC, AAD ($p$ = 0.006), *TERT* promoter mutation status ($p < 0.001$), and RAI total dose ($p$ = 0.015) were significant. In an analysis including the interaction between AAD and *TERT* promoter mutation (AAD ≥ 55 and mutant *TERT*, AGE×*TERT*) as one of the variables, AGE×*TERT* ($p$ = 0.005) was significant in addition to AAD ($p < 0.001$), *TERT* promoter mutation ($p < 0.001$), and RAI total dose ($p$ = 0.034) (Table 2). This means that the effect of AAD on mortality rate was different according to *TERT* promoter mutation status.

## Comparison of adjusted hazard ratios (HRs) of age at diagnosis according to TERT promoter mutation status

A subgroup analysis of DTC patients according to *TERT* promoter mutation was conducted using multivariable Cox analysis. In the WT-*TERT* group, the 10-year survival rate was 99.3%

**Table 2. The association between clinicopathological variables, including age-*TERT* interaction and cancer-specific survival (CSS), in patients with differentiated thyroid cancer (DTC).**

| Variables | Total DTC (*n* = 393) | | | | No interaction | | | | Interaction | | | |
|---|---|---|---|---|---|---|---|---|---|---|---|---|
| | Death | | N | 10-year survival rate (%) | Multivariate Cox models | | | | Multivariate Cox models | | | |
| | No | Yes | | | Hazard ratio | 95% lower | 95% upper | *p* value | Hazard ratio | 95% lower | 95% upper | *p* value |
| Sex | | | | | | | | | | | | |
| Female | 309 | 20 | 329 | 95.1 | 1.00 (reference) | | | 0.123 | 1.00 (reference) | | | 0.086 |
| Male | 57 | 7 | 64 | 92.2 | 2.08 | 0.82 | 5.29 | | 2.25 | 0.89 | 5.67 | |
| Age at diagnosis (years) | | | | | | | | | | | | |
| <55 | 311 | 8 | 319 | 97.5 | 1.00 (reference) | | | 0.006 | 1.00 (reference) | | | <0.001 |
| ≥55 | 55 | 19 | 74 | 82.4 | 3.88 | 1.48 | 10.19 | | 18.94 | 4.12 | 87.03 | |
| *TERT* promoter mutations | | | | | | | | | | | | |
| WT | 342 | 8 | 350 | 98.0 | 1.00 (reference) | | | <0.001 | 1.00 (reference) | | | <0.001 |
| Mutation | 24 | 19 | 43 | 67.4 | 6.83 | 2.55 | 18.31 | | 38.78 | 7.79 | 192.98 | |
| AGE * TERT | | | | | | | | | | | | |
| Else | | | | | | | | | 1.00 (reference) | | | 0.005 |
| Age at diagnosis ≥55 and *TERT* mutation | | | | | | | | | 0.07 | 0.01 | 0.44 | |
| Histological type | | | | | | | | | | | | |
| PTC | 310 | 17 | 327 | 96.6 | 1.00 (reference) | | | 0.053 | 1.00 (reference) | | | 0.084 |
| FTC | 56 | 10 | 66 | 84.8 | 2.62 | 0.99 | 6.93 | | 2.39 | 0.89 | 6.41 | |
| Extrathyroidal extension | | | | | | | | | | | | |
| Absent | 334 | 18 | 352 | 96.0 | 1.00 (reference) | | | 0.368 | 1.00 (reference) | | | 0.413 |
| Present | 32 | 9 | 41 | 82.9 | 1.55 | 0.60 | 4.01 | | 1.46 | 0.59 | 3.65 | |
| Distant metastasis | | | | | | | | | | | | |
| Absent | 352 | 18 | 370 | 96.8 | 1.00 (reference) | | | 0.133 | 1.00 (reference) | | | 0.133 |
| Present | 14 | 9 | 23 | 60.9 | 2.27 | 0.78 | 6.57 | | 2.28 | 0.78 | 6.69 | |
| Tumor size | | | | | | | | | | | | |
| < 2.0 cm | 43 | 2 | 45 | 95.6 | 1.00 (reference) | | | 0.799 | 1.00 (reference) | | | 0.621 |
| 2.0–4.0 cm | 277 | 16 | 293 | 95.9 | 0.87 | 0.21 | 3.54 | 0.846 | 1.32 | 0.31 | 5.65 | 0.707 |
| > 4.0 cm | 46 | 9 | 55 | 87.3 | 1.22 | 0.24 | 6.09 | 0.811 | 2.02 | 0.39 | 10.50 | 0.404 |
| RAI total dose | | | | | | | | | | | | |
| Per 1 mCi | | | 393 | | 1.0015 | 1.0003 | 1.0027 | 0.015 | 1.0013 | 1.0001 | 1.0024 | 0.034 |

Abbreviations: *TERT*, telomerase reverse transcriptase; PTC, papillary thyroid cancer; FTC, follicular thyroid cancer; AJCC, American Joint Committee on Cancer; RAI, radioactive iodine.

**Table 3. A comparison of the adjusted hazard ratio (HR) of age at diagnosis in the wild-type *TERT* (WT-*TERT*) and mutant *TERT* (M-*TERT*) groups of differentiated thyroid cancer (DTC).**

| | Subgroup with TERT | | | | | | | | | | | | | | | |
| --- | --- | --- | --- | --- | --- | --- | --- | --- | --- | --- | --- | --- | --- | --- | --- | --- |
| | **TERT mutant** | | | | | | | | **TERT wild-type** | | | | | | | |
| | Death | | N | 10-year survival rate (%) | Multivariate Cox models | | | | Death | | N | 10-year survival rate (%) | Multivariate Cox models | | | |
| | 0 | 1 | | | Hazard ratio | 95% lower | 95% upper | *p* value | 0 | 1 | | | Hazard ratio | 95% lower | 95% upper | *p* value |
| **Sex** | | | | | | | | | | | | | | | | |
| Female | 21 | 15 | 36 | 69.4 | 1.00 (reference) | | | 0.455 | 288 | 5 | 293 | 98.3 | 1.00 (reference) | | | 0.769 |
| Male | 3 | 4 | 7 | 57.1 | 1.61 | 0.46 | 5.61 | | 54 | 3 | 57 | 96.5 | 1.38 | 0.16 | 11.64 | |
| **Age at diagnosis (years)** | | | | | | | | | | | | | | | | |
| < 55 | 9 | 6 | 15 | 60.0 | 1.00 (reference) | | | 0.894 | 302 | 2 | 304 | 99.3 | 1.00 (reference) | | | <0.001 |
| ≥ 55 | 15 | 13 | 28 | 71.4 | 1.07 | 0.38 | 3.05 | | 40 | 6 | 46 | 89.1 | 20.78 | 3.89 | 110.97 | |
| **Histological type** | | | | | | | | | | | | | | | | |
| PTC | 19 | 13 | 32 | 75.0 | 1.00 (reference) | | | 0.377 | 291 | 4 | 295 | 99.0 | 1.00 (reference) | | | 0.095 |
| FTC | 5 | 6 | 11 | 45.5 | 1.90 | 0.46 | 7.86 | | 51 | 4 | 55 | 92.7 | 3.71 | 0.80 | 17.28 | |
| **Extrathyroidal extension** | | | | | | | | | | | | | | | | |
| Absent | 20 | 10 | 30 | 76.7 | 1.00 (reference) | | | 0.125 | 314 | 8 | 322 | 97.8 | 1.00 (reference) | | | 0.376 |
| Present | 4 | 9 | 13 | 46.2 | 2.31 | 0.79 | 6.75 | | 28 | 0 | 28 | 100.0 | 0.21 | 0.01 | 6.62 | |
| **Distant metastasis** | | | | | | | | | | | | | | | | |
| Absent | 22 | 13 | 35 | 77.1 | 1.00 (reference) | | | 0.315 | 330 | 5 | 335 | 98.8 | 1.00 (reference) | | | 0.374 |
| Present | 2 | 6 | 8 | 25.0 | 2.07 | 0.50 | 8.55 | | 12 | 3 | 15 | 80.0 | 2.68 | 0.30 | 23.51 | |
| **Tumor size** | | | | | | | | | | | | | | | | |
| < 2.0 cm | 3 | 1 | 4 | 75.0 | 1.00 (reference) | | | 0.741 | 40 | 1 | 41 | 97.6 | 1.00 (reference) | | | 0.323 |
| 2.0–4.0 cm | 19 | 11 | 30 | 76.7 | 1.51 | 0.21 | 10.77 | 0.684 | 258 | 5 | 263 | 98.1 | 0.95 | 0.10 | 9.29 | 0.965 |
| > 4.0 cm | 2 | 7 | 9 | 33.3 | 2.26 | 0.23 | 21.73 | 0.481 | 44 | 2 | 46 | 97.8 | 4.86 | 0.24 | 98.75 | 0.304 |
| **RAI total dose** | | | | | | | | | | | | | | | | |
| Per 1 mCi | | | 43 | | 1.0011 | 0.9997 | 1.0025 | 0.123 | | | 350 | | 1.0027 | 0.9993 | 1.0062 | 0.123 |

Abbreviations: *TERT*, telomerase reverse transcriptase; PTC, papillary thyroid cancer; FTC, follicular thyroid cancer; AJCC, American Joint Committee on Cancer; RAI, radioactive iodine.

in patients < 55 years old and 89.1% in patients ≥ 55 years. The adjusted HR of AAD ≥ 55 years was 20.78 (95% CI: 3.89–110.97; $p < 0.001$). The risk of AAD ≥ 55 on the mortality rate in the WT-*TERT* group was consistent with the results of the multivariable analysis of the total DTC group. In contrast, the risk of AAD ≥ 55 was not significant in the M-*TERT* group, with adjusted HR of 1.07 (95% CI: 0.38–3.05; $p = 0.894$) (Table 3).

The adjusted HR plots in Fig 4 show the associations between AAD as a continuous variable and mortality rate. A linear association was observed between patient AAD and mortality rate in all patients with DTC and in the WT-*TERT* group (Fig 4A and 4C). Plots of the M-*TERT* group showed a trend of increasing adjusted HR with AAD when the patient was older than 55 years, but without significance (Fig 4B).

## Mediation analysis

The results of regression analyses in mediation analysis are presented as HRs (Table 4). PDE, the direct effect of AAD on mortality without mediation, showed an increased risk with significance (HR: 8.55; 95% CI: 1.04–51.88; $p = 0.045$). PIE, the effect of AAD on mortality only via mediation by *TERT* promoter mutation, also showed an increased risk with significance (HR:

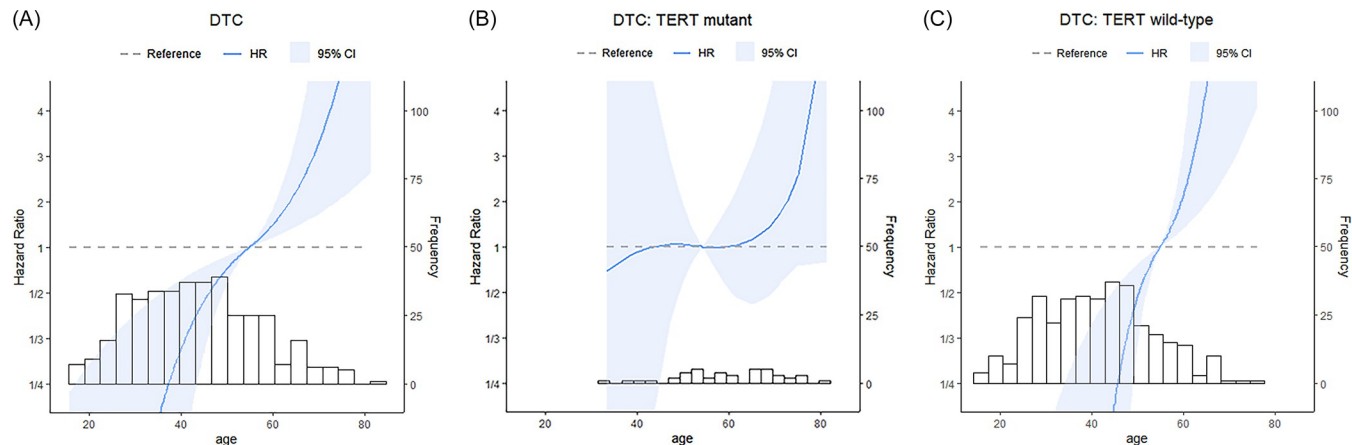

**Fig 4. Multivariable Cox regression analysis of cancer-specific mortality risk as adjusted hazard ratio (HR) plots.** The X-axis represents age at diagnosis (AAD) as a continuous variable, left Y-axis represents adjusted hazard ratio (HR) of mortality rate, and right Y-axis represents number of patients included in each AAD groups. Blue line represents HR, and the range in light blue represents the 95% confidence interval. (A) HR plot of total patients with differentiated thyroid cancer, (B) HR plot of mutant *TERT* (M-*TERT*), and (C) HR plot of wild-type *TERT* (WT-*TERT*).

5.80; 95% CI: 1.95–14.10; $p = 0.001$). INT_med, the alteration of AAD effect on mortality by *TERT* promoter mutation, showed a decreased risk (HR: 0.26; 95% CI: 0.08–0.75; $p = 0.014$), indicating that the effect of AAD on mortality is undermined in the presence of *TERT* promoter mutation via its interaction with AAD. TE, the sum of PDE, PIE, and INT_med, showed an HR of 12.85 with significance (HR: 12.85; 95% CI: 1.62–58.20; $p = 0.013$).

Subsequently, the proportion_mediated was calculated using the PDE and TIE with the formula PDE (TIE—1)/(PDE x TIE -1) as 36%, which indicates that a *TERT* promoter mutation explains 36% of the effects of AAD and mortality rate in patients with DTC.

A sensitivity analysis was used to assess the HRs of PDE and PIE in several AAD cutoffs as 45, 55, and 65, and the estimators were quite consistent (Fig 5). The mediational E-values were 11.08 and 5.21 for the PIE estimate and the limit of its 95% CI closest to the null, respectively. To completely explain away the observed PIE HR of 5.8, an unmeasured confounder associated with both the *TERT* and overall survival with an approximate HR of 11.08-fold each, above and beyond the measured covariates, could suffice, but weaker confounding could not.

## Discussion

The purpose of this study was to explore the role of *TERT* promoter mutation in age-associated mortality in patients with DTC. A significant interaction between AAD and *TERT* promoter mutation on mortality rate was confirmed in a multivariable Cox regression analysis.

**Table 4. Regression analyses in mediation analysis.**

|  | Hazard ratio | 95% Lower | 95% Upper | *p* value |
|---|---|---|---|---|
| Total effect (TE) | 12.85 | 1.62 | 58.2 | 0.013 |
| Pure direct effect (PDE) | 8.55 | 1.04 | 51.88 | 0.045 |
| Pure indirect effect (PIE) | 5.8 | 1.95 | 14.1 | 0.001 |
| Mediated interaction (INT_med) | 0.26 | 0.08 | 0.75 | 0.014 |
| Total indirect effect (TIE) | 1.5 | 0.62 | 2.83 | 0.477 |
| % Mediated | 36 |  |  |  |

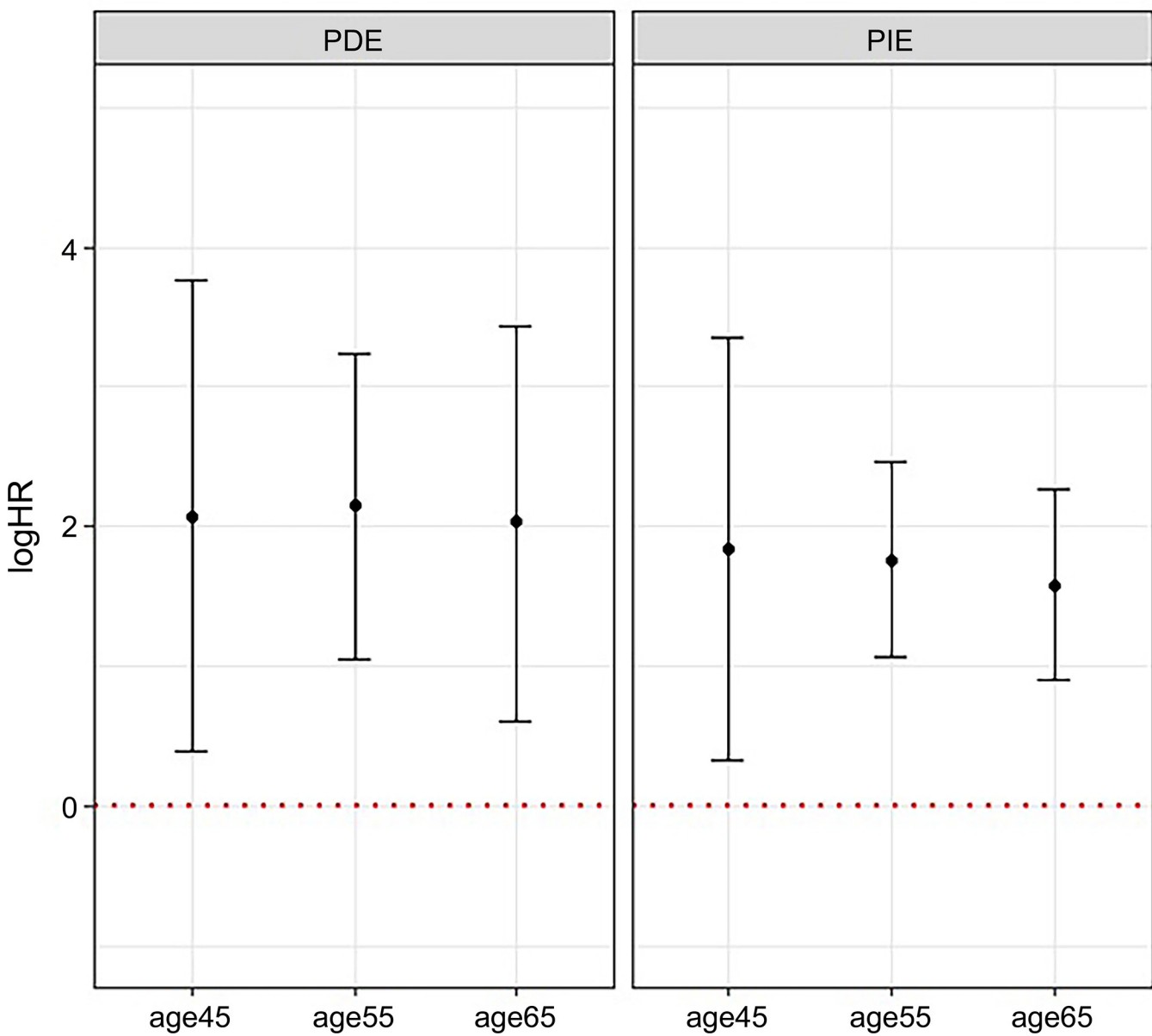

**Fig 5. Hazard ratios of the pure direct effect (PDE) and the pure indirect effect (PIE) at several cutoffs of age at diagnosis.** Sensitivity analyses performed on pure direct effect (PDE) and pure indirect effect (PIE). The X-axis represents different age at diagnosis (AAD) cutoffs, and the Y-axis represent log hazard ratio (logHR). LogHRs showed consistent value when applying different AAD cutoffs (45, 55, and 65).

Furthermore, we found that the effect of AAD on the mortality rate was partially mediated by *TERT* promoter mutation.

In multivariable Cox regression analysis with interaction model, AAD $\geq$ 55 and *TERT* promoter mutation showed significantly increased HRs, revalidating the results of previous studies that have emphasized the importance of *TERT* promoter mutation as a robust prognostic marker in thyroid cancers [11–14, 33–36].

A significant interaction between AAD and *TERT* promoter mutation was correlated with the non-linear association of AAD and mortality rate in M-*TERT* group. WT-*TERT* group in which an interaction between AAD and *TERT* promoter mutation is absent showed linear

pattern of age-associated mortality rate. In addition, we hypothesized that age-associated mortality consists of three components as direct effect of AAD, pure indirect effect through *TERT* promoter mutation, and the interaction between AAD and *TERT* promoter mutation, and conducted a mediation analysis to verify it. The mediation analysis revealed that in patients with DTC, *TERT* promoter mutation contributes to age-associated mortality by 36% through pure indirect effect (PDE) and mediated interaction (INT_med).

In cases of multifactorial disease such as cancers, as the complex causality of the disease unfolds, identification of an interaction between the causal factors might help clinicians to better understand the pathophysiology; Spiegl-Kreinecker et al. found a complex interaction between the rs2853669 polymorphism and the *TERT* promoter mutation status in patients with glioblastoma [37], while Vuong et al. demonstrated an interaction between *TERT* and $O^6$-methylguanine-DNA methyltransferase in patients with glioma [38].

Additionally, the C228T and C250T are two dominant hotspots of *TERT* promoter mutations found in many cancers including thyroid cancer, melanoma, urothelial cancer, and glioblastoma [10, 39, 40]. The C228T is much more frequent subtype compared to the C250T in many cancers except melanoma [41, 42]. The prognostic role of C228T and C250T subtypes have been collectively investigated across various studies. However, it is not well established which mutation harbors worse prognosis. In melanoma, conflicting results have been presented. According to Andrés-Lencina et al. and Chang et al., C228T harbored poor outcome than C250T [43, 44]. On the contrary, Del Bianco et al. suggested that C250T was associated with poor survival compared to C228T [45]. In a study of 358 glioblastoma patients, Nonoguchi et al. revealed that only C228T subtype was significantly showed worse prognosis [46]. Further investigations are required to elucidate different prognostic role of the two dominant types of *TERT* promoter mutation, especially in research on thyroid cancer [47, 48].

To our knowledge, this study is the first to investigate the interaction between AAD and *TERT* promoter mutation in patients with DTC and notably demonstrated the mediation effect of *TERT* promoter mutation on the age-associated mortality rate in DTC. The results of our study support the validity of a study that incorporated *TERT* promoter mutation into the WHO 2017 classification of FTC [17], and another study that proposed a new prognostic system incorporating *TERT* promoter mutation status into the TNM-8 [18]. This study further emphasized the importance of *TERT* promoter mutation as an essential nonanatomical prognostic factor in thyroid cancer.

There are several limitations of this study. First, there is an inherent drawback of any mediation analysis, which is based on the hypothesis that there are no unmeasured confounders in the causal relationships between exposure, mediator, and outcome. However, we cannot exclude the possibility of unmeasured confounders. Second, this study was conducted retrospectively at a single tertiary medical center resulting in small numbers of mutant *TERT* patients and lack of validation. Third, the analyses were performed with DTC by combining the PTC and FTC, and it was underpowered to perform a subgroup analysis for each subtype. However, considering that PTC is the most common subtype among DTC and that *TERT* promoter mutation acts as a prognostic marker in FTC [17, 49, 50] further analysis with a larger subset of PTC and FTC patients might produce a relevant result regarding the mediation effect of *TERT* promoter mutation on cancer-specific mortality. Finally, mortality pattern varied by gender in both WT-*TERT* and M-*TERT*, however, meaningful comparative analysis of mortality by AAD and gender subgroups was not possible due to limited size of cohort. In male, the mortality rate increased with AAD in WT-*TERT*, and decreased in M-*TERT*. Unlikely, in female, the mortality rate was highest in the 55–64 AAD group, and lower in the ≥65 AAD group than in the 55–64 AAD group in WT-*TERT*. In M-*TERT*, "Z-shaped" pattern similar to the pattern of M-*TERT* in Fig 2 was observed. Interestingly, a study by Grasselli et al. showed

an association between estradiol levels and TERT activity in human endothelial cells [51]. Further research with a larger cohort and investigation of the interaction of AAD, gender and *TERT* promoter mutations would be worthwhile.

## Conclusion

The findings in our study highlights the robust role of *TERT* promoter mutation in age-associated mortality in DTC by mediation analysis. Risk of cancer-specific death in DTC by AAD, which is a well-established risk factor, is partially explicable by *TERT* promoter mutation by 36%. Considering this non-negligible percentage, it is necessary to build a consensus on staging system that takes both AAD and *TERT* promoter mutation into consideration. Further research to explore how *TERT* promoter mutation can be appropriately integrated into forthcoming staging system, for instance, applying different AAD cutoffs for WT-*TERT* and M-*TERT*, is needed.

## Supporting information

**S1 Table. The univariable analysis of the association between clinicopathological variables and cancer-specific survival (CSS) in patients with differentiated thyroid cancer.** (DOCX)

**S2 Table. The univariable analysis of the association between clinicopathological variables and cancer-specific survival (CSS) in patients with papillary thyroid cancer (PTC).** (DOCX)

## Author Contributions

**Conceptualization:** Jung Heo, Sungjoo Lee, Kyunga Kim, Tae Hyuk Kim.

**Data curation:** Jung Heo, Jun Park, Tae Hyuk Kim.

**Formal analysis:** Sungjoo Lee, Kyunga Kim.

**Investigation:** Chang-Seok Ki.

**Supervision:** Kyunga Kim, Tae Hyuk Kim.

**Validation:** Sun Wook Kim, Jae Hoon Chung.

**Visualization:** Sungjoo Lee, Kyunga Kim.

**Writing – original draft:** Jung Heo.

**Writing – review & editing:** Jung Heo, Sungjoo Lee, Heera Yang, Hyunju Park, Young Lyun Oh, Hye In Kim, Sun Wook Kim, Jae Hoon Chung, Kyunga Kim, Tae Hyuk Kim.

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
