## [Decision Letter · Decision Letter 0]

18 Jul 2023

PONE-D-23-13935Age-associated mortality is partially mediated by TERT promoter mutation status in differentiated thyroid carcinomaPLOS ONE

Dear Dr. Kim,

Thank you for submitting your manuscript to PLOS ONE. After careful consideration, we feel that it has merit but does not fully meet PLOS ONE’s publication criteria as it currently stands. Therefore, we invite you to submit a revised version of the manuscript that addresses the points raised during the review process.

We look forward to receiving your revised manuscript.

Kind regards,

Avaniyapuram Kannan Murugan, M.Phil., Ph.D.

Academic Editor

PLOS ONE

Journal Requirements:

Additional Editor Comments:

Some potential articles in this field are not cited and are worth to cite the following articles as they contributed to the development of this field (PMID: 23766237; PMID: 25024077; PMID: 26354077; PMID: 26711586; PMID: 26902827).

Reviewers' comments:

Reviewer's Responses to Questions

**Comments to the Author**

1. Is the manuscript technically sound, and do the data support the conclusions?

Reviewer #1: Partly

Reviewer #2: Partly

Reviewer #3: Yes

2. Has the statistical analysis been performed appropriately and rigorously? 

Reviewer #1: Yes

Reviewer #2: N/A

Reviewer #3: Yes

3. Have the authors made all data underlying the findings in their manuscript fully available?

Reviewer #1: Yes

Reviewer #2: Yes

Reviewer #3: Yes

4. Is the manuscript presented in an intelligible fashion and written in standard English?

Reviewer #1: No

Reviewer #2: Yes

Reviewer #3: Yes

5. Review Comments to the Author

Reviewer #1: Age-associated mortality is partially mediated by TERT promoter mutation status in differentiated thyroid carcinoma

[Overall]

This study examined the relationship between age, TERT promoter mutations, and mortality in differentiated thyroid carcinoma (DTC). The findings revealed that age was associated with increased mortality in all DTC patients and those without TERT promoter mutations. A significant interaction was identified between age and TERT promoter mutation in relation to the mortality rate in DTC. Furthermore, this study revealed that the impact of age on the mortality rate in DTC was partially influenced by the presence of TERT promoter mutation using multiple analysis. The authors emphasize the influence of TERT promoter mutation on cancer-specific outcomes based on these analyses. They also highlight the importance of incorporating this critical nonanatomical variable into future stage classifications to enhance the development of personalized prediction models for thyroid cancer.

[Critique]

Overall, the authors addressed evidence about the partial correlation between morality rate and TERT promoter mutation, showing that TERT promoter mutation contributes to age-associated mortality by 36% in TERT mutant DTC. This interaction was theoretically considered as factor of poor outcome in DTC patients with older age and this paper confirmed the theory by analyzing patient samples. This work is logically addressed and appropriately presented in general, but the following points should be addressed or discussed before further consideration for the publication. Especially, many of the details about the figures are missing or incomplete.

[Major comments]

1. Many of the figure legends are missing or have lack of the information. For example, figure 2, 3, 4, and 5 have only title and there are no figure legend which can explain about detail of the figure. It would be better to have abbreviation, statistic information, explanation about each subfigure (such as A, B, and C), etc. in the legend. Especially, in the case of dual axis graph, like figure 4, detailed figure legend should be needed to prevent confusion.

2. In figure 2 and related manuscript, the authors addressed that “The mortality rates were low in all patients with DTC and in the WT-TERT group before the age of 55 years compared with the high mortality rate in the M-TERT group”. To conclude more precisely, the author need to add statistical analysis with comparison between “all vs M-TERT” and “WT-TERT vs M-TERT” and provide information in the figure 2.

3. 2. Based on the mutation analysis, the authors could see two different type of promoter mutations, C228T and C250T. Have there been any previous studies conducted on which mutation is a more risk factor or more frequent in diseases other than DTC (for example, glioblastoma multiforme, GBM)? If so, it would be great to do more discussion in addition to line 429-434.

[Minor comments]

1. In Table 4 and related manuscript, the authors showing result of mediation analysis. Even though they mentioned in the introduction part (line 272) that “natural direct effect (NDE), a synonym for PDE, and the natural indirect effect (NIE), a synonym for TIE”, it would be great to understand for the reader using unified term. In addition, please add abbreviation as well in the table; for example, Total effect (TE), Pure direct effect (PDE), etc. It makes the easier to follow.

Reviewer #2: The manuscript submitted by Jung Heo et al., is a retrospective study that examines the interaction between age and TERT promoter mutation in 393 patients with differentiated thyroid carcinoma and its effect on mortality. The authors subtracted DNA from FFPE tissue and identified TERT promoter mutations and mutant enrichment in those patients. The authors then applied a series of statistical analyses including multivariable Cox regression and mediation analysis to investigate the weight of various factors such as age, treatment and TERT promoter mutation on the survival rate of this group of patients. The examination of interaction between age and TERT promoter mutation is appraisable and the results are interesting.

In addition to the drawbacks of the study mentioned by the authors in the discussion part, I have few more comments about the data analysis and presentation.

1. Most of the patients in this cohort has papillary thyroid cancer, which occurs more frequently in women. It seems that gender is an unneglectable factor associated to the differentiated thyroid carcinoma.

In table 1 and S1, the authors listed the numbers of females and males of the patient cohort and the number of patients with TERT promoter mutations. However, they did not specify the number of female and male in the M-TERT group. Is there a difference in the incidence of TERT promoter mutations between males and females? Moreover, what is the incidence of TERT promoter mutations in different age groups?

As presented in figures 2, 3B, and 3E, there is a clear linear association between mortality rate and age in patients with WT-TERT promoters, whereas this linear association is not found in patients with mutated TERT promoters. The mortality rate is lower in the 55-64y age group as compared to that in the 45-54y age group in patients with mutated TERT promoters. Have the authors examined the mortality rate separately in males and females in different age groups? Females go through menopause between 45 and 55, and there is a decline of estradiol production during this period. It has been reported that 17beta-estradiol significantly increased telomerase activity in human endothelial cells (Grasselli et al., Circ Res. 2008 Jul 3;103(1):34-42), therefore, it is worthwhile and crucial to explore the interaction of age, gender and TERT promoter mutations in this cohort of patients.

2. Figure 3:

To make it clear and easier for reading, please put subtitles to each graph to indicate the group of patients.

Reviewer #3: Aging and TERT promoter mutations have been previously shown as prognostic factors in Differentiated Thyroid Carcinoma (Tae Hyuk Kim et al., 2016). In this study, authors used their previous Differentiated Thyroid Carcinoma (DTC) cohort to elucidate the interaction of aging and TERT promoter mutations. They found that the effect of aging on mortality of mutant TERT patients were not significant, while aging caused higher mortality rate in Wild type TERT patients. In addition, Heo J et al., found that TERT promoter mutation contributes to 36% of age-related mortality rate in this DTC cohort. Next, authors concluded that TERT promoter mutations should be considered in a stage classification of DTC. Although these finding are interesting observations, but the conclusion of this study is similar to their previous study (Tae Hyuk Kim et al., 2016).

6. PLOS authors have the option to publish the peer review history of their article (what does this mean?). If published, this will include your full peer review and any attached files.

Reviewer #1: No

Reviewer #2: No

Reviewer #3: No

---

## [Author Response · Author response to Decision Letter 0]

13 Sep 2023

[Response to Reviewers’ comments]

Manuscript Number: PONE-D-23-13935

Article Type: Original article

Title: Age-associated mortality is partially mediated by TERT promoter mutation status in differentiated thyroid carcinoma

[Additional Editor Comments]

Some potential articles in this field are not cited and are worth to cite the following articles as they contributed to the development of this field (PMID: 23766237; PMID: 25024077; PMID: 26354077; PMID: 26711586; PMID: 26902827).

* Response: We appreciate the editor’s introduction of great articles that we overlooked for citation. We have added suitable articles to the references (Ref 7–9, 46–47). 

[Reviewer #1]

[Major comments]

1) Many of the figure legends are missing or have lack of the information. For example, figure 2, 3, 4, and 5 have only title and there are no figure legend which can explain about detail of the figure. It would be better to have abbreviation, statistic information, explanation about each subfigure (such as A, B, and C), etc. in the legend. Especially, in the case of dual axis graph, like figure 4, detailed figure legend should be needed to prevent confusion.

* Response: Thank you for the valuable comment. We have enhanced the figure legends by adding detailed information. 

* [Revised] Line 198-203, 239–247, 258–266, 311–316, 344–347.

2) In figure 2 and related manuscript, the authors addressed that “The mortality rates were low in all patients with DTC and in the WT-TERT group before the age of 55 years compared with the high mortality rate in the M-TERT group”. To conclude more precisely, the author need to add statistical analysis with comparison between “all vs M-TERT” and “WTTERT vs M-TERT” and provide information in the figure 2.

* Response: According to the comment, we performed Cochran-Mantel-Haenszel (CMH) test to compare the mortality rate between WT-TERT and M-TERT groups, and the mortality rate was higher in M-TERT than WT-TERT (p=0.005). We added this result in the revised manuscript regarding Fig 2. 

As for “All vs. M-TERT”, concerning that “All” encompasses “M-TERT”, the phrase about “All vs. M-TERT” was removed from the original text, although M-TERT was significantly associated with shorter cancer-specific survival compared to both all patients with DTC and WT-TERT (log-rank test p<0.001). 

* [Revised] Line 147–149, 231–233 

3) Based on the mutation analysis, the authors could see two different type of promoter mutations, C228T and C250T. Have there been any previous studies conducted on which mutation is a more risk factor or more frequent in diseases other than DTC (for example, glioblastoma multiforme, GBM)? If so, it would be great to do more discussion in addition to line 429-434.

* Response: Thank you for the valuable comment. We added a paragraph discussing previous studies on the difference between C228T and C250T as prognostic factors in cancers other than thyroid cancer. 

* [Revised] Line 375–386 

[Minor comments]

1) In Table 4 and related manuscript, the authors showing result of mediation analysis. Even though they mentioned in the introduction part (line 272) that “natural direct effect (NDE), a synonym for PDE, and the natural indirect effect (NIE), a synonym for TIE”, it would be great to understand for the reader using unified term. In addition, please add abbreviation as well in the table; for example, Total effect (TE), Pure direct effect (PDE), etc. It makes the easier to follow.

* Response: Thank you for the opinion. We removed the potentially confusing terms (NDE and NIE) and unified the terminology. Also, we added abbreviations in the Table 4.

* [Revised] Line 333–334 and 1st column of Table 4

[Reviewer #2]

1) Most of the patients in this cohort has papillary thyroid cancer, which occurs more frequently in women. It seems that gender is an unneglectable factor associated to the differentiated thyroid carcinoma. In table 1 and S1, the authors listed the numbers of females and males of the patient cohort and the number of patients with TERT promoter mutations. However, they did not specify the number of female and male in the M-TERT group. Is there a difference in the incidence of TERT promoter mutations between males and females?

* Response: In DTC, the incidence of TERT promoter mutation were 10.9% (36/329) in female, and 10.9% (7/64) in male, respectively (p=1.000). In PTC, the incidence of the mutation were 9.1% (25/276) in female, and 13.7% (7/51) in male, respectively (p=0.214).

Moreover, what is the incidence of TERT promoter mutations in different age groups?

* Response: The incidence of TERT promoter mutation in each age groups are 1.8% (4/224) in age<45, 11.6% (11/95) in age 45–54, 23.3% (10/43) in age 55–64, and 58.1% (18/31) in age≥65, respectively. 

As presented in figures 2, 3B, and 3E, there is a clear linear association between mortality rate and age in patients with WT-TERT promoters, whereas this linear association is not found in patients with mutated TERT promoters. The mortality rate is lower in the 55-64y age group as compared to that in the 45-54y age group in patients with mutated TERT promoters. Have the authors examined the mortality rate separately in males and females in different age groups? 

* Response: As shown in the additional Fig 1 and 2, mortality rate increased with age in total patients with DTC in both male and female subgroups (black curves). In M-TERT, the mortality rate decreased with age in males (additional Fig 1), and the mortality rate showed a “Z shape” in females (additional Fig 2) (p for trend; 0.147 in males and 0.054 in females) (red curves). Of note, the graph of mortality rate in total M-TERT is similar to that of female M-TERT. This may be due to the dominance of females over males, which is approximately 5:1 in the study cohort. In WT-TERT, the mortality rate increased with age in the male subgroup (p for trend=0.002), whereas that of the female subgroup was highest in the 55-64 age group, and lower in the > 65 age group than in the 55-64 age group (p for trend=0.0001) (yellow curves). 

We appreciate the reviewer’s suggestion for advanced analyses of age- and gender-specific mortality rates. Mortality patterns varied by gender in both WT-TERT and M-TERT, as noted above. However, we regret to inform the reviewer that the number of deaths in each age and gender subgroup was too small, particularly with only seven total deaths in the entire male group, to allow meaningful comparative analysis of mortality by age and gender subgroups. 

Females go through menopause between 45 and 55, and there is a decline of estradiol production during this period. It has been reported that 17beta-estradiol significantly increased telomerase activity in human endothelial cells (Grasselli et al., Circ Res. 2008 Jul 3;103(1):34-42), therefore, it is worthwhile and crucial to explore the interaction of age, gender and TERT promoter mutations in this cohort of patients.

* Response: 

The authors understand the reviewer’s point regarding the need to investigate the effect of menopause and changes in E2 levels on TERT activity in female patients. Despite the intriguing perspective, there was no obvious pattern observed in mortality rates around the age of 45-54 in the female group, especially within the WT-TERT where the influence of TERT promoter mutation was excluded, suggesting that E2-dependent induction of hTERT would be a cell type specific event requiring NO production. Thus, a better understanding of the TERT activity based on estradiol levels may be pursued in the field of thyroid cancer. We have mentioned this point in the Discussion section.

* [Revised] Line 406–410.

2) To make it clear and easier for reading, please put subtitles to each graph to indicate the group of patients.

* Response: Thank you for the valuable comment. We have enhanced the figure legends by adding detailed information explaining each graph. 

* [Revised] Line 198-203, 239–247, 258–266, 311–316, 344–347.

[Reviewer #3]

Aging and TERT promoter mutations have been previously shown as prognostic factors in Differentiated Thyroid Carcinoma (Tae Hyuk Kim et al., 2016). In this study, authors used their previous Differentiated Thyroid Carcinoma (DTC) cohort to elucidate the interaction of aging and TERT promoter mutations. They found that the effect of aging on mortality of mutant TERT patients were not significant, while aging caused higher mortality rate in Wild type TERT patients. In addition, Heo J et al., found that TERT promoter mutation contributes to 36% of age-related mortality rate in this DTC cohort. Next, authors concluded that TERT promoter mutations should be considered in a stage classification of DTC. Although these finding are interesting observations, but the conclusion of this study is similar to their previous study (Tae Hyuk Kim et al., 2016).

* Response: Thank you for the comment, and we agree with the reviewer’s opinion that conclusion of this study is similar to previous studies. What is different from previous studies is that this study found TERT promoter mutation contributes partially to the prognostic role of age at diagnosis in DTC. To emphasize this distinguishing factor, we partially modified the manuscript mentioning the need for further research on how to incorporate TERT with age into a new staging system (for example, different age cutoffs based on TERT promoter mutation status). 

* [Revised] Line 414-421

---

## [Decision Letter · Decision Letter 1]

3 Oct 2023

PONE-D-23-13935R1Age-associated mortality is partially mediated by TERT promoter mutation status in differentiated thyroid carcinomaPLOS ONE

Dear Dr. Kim,

Thank you for submitting your manuscript to PLOS ONE. After careful consideration, we feel that it has merit but does not fully meet PLOS ONE’s publication criteria as it currently stands. Therefore, we invite you to submit a revised version of the manuscript that addresses the points raised during the review process.

We look forward to receiving your revised manuscript.

Kind regards,

Avaniyapuram Kannan Murugan, M.Phil., Ph.D.

Academic Editor

PLOS ONE

Journal Requirements:

Additional Editor Comments:

Kindly address the pending reviewer comments which has merit and to be addressed before acceptance.

Reviewers' comments:

Reviewer's Responses to Questions

**Comments to the Author**

1. If the authors have adequately addressed your comments raised in a previous round of review and you feel that this manuscript is now acceptable for publication, you may indicate that here to bypass the “Comments to the Author” section, enter your conflict of interest statement in the “Confidential to Editor” section, and submit your "Accept" recommendation.

Reviewer #1: All comments have been addressed

Reviewer #2: (No Response)

Reviewer #3: All comments have been addressed

2. Is the manuscript technically sound, and do the data support the conclusions?

Reviewer #1: Yes

Reviewer #2: Partly

Reviewer #3: Yes

3. Has the statistical analysis been performed appropriately and rigorously? 

Reviewer #1: Yes

Reviewer #2: Yes

Reviewer #3: Yes

4. Have the authors made all data underlying the findings in their manuscript fully available?

Reviewer #1: Yes

Reviewer #2: No

Reviewer #3: Yes

5. Is the manuscript presented in an intelligible fashion and written in standard English?

Reviewer #1: Yes

Reviewer #2: Yes

Reviewer #3: Yes

6. Review Comments to the Author

Reviewer #1: In this revised iteration, the authors have conscientiously addressed the majority of the questions that I had raised. The current manuscript has been rendered more lucid and accessible, thereby facilitating a better understanding of its content. It is worth noting that in addition to the modifications to the advice I suggested, additional modifications allowed the author to further emphasize differences in relation to existing research results. Overall, it is evident that this manuscript has undergone substantial enhancements compared to its original submission. These improvements confirmed previously published research results and suggested additional direction to academics and readers by suggesting the need for additional research in elucidating the relationship between aging, TERT promoter mutation, and DTC diagnosis. As the author mentioned, there are limitations such as the possibility of unmeasured confounders and a small cohort, but the limitations and the need for additional research are appropriately described.

Reviewer #2: (No Response)

Reviewer #3: (No Response)

7. PLOS authors have the option to publish the peer review history of their article (what does this mean?). If published, this will include your full peer review and any attached files.

Reviewer #1: No

Reviewer #2: No

Reviewer #3: No

---

## [Author Response · Author response to Decision Letter 1]

25 Oct 2023

[Response to Reviewers’ comments]

Manuscript Number: PONE-D-23-13935R1

Article Type: Original article

Title: Age-associated mortality is partially mediated by TERT promoter mutation status in differentiated thyroid carcinoma

[Journal Requirements]

Any changes to the reference list should be mentioned in the rebuttal letter that accompanies your revised manuscript.

- Response: A new article is now included in the references as reference number 26 to provide an explanation for the abbreviation ‘AAD’ (age at diagnosis). Apart from this addition, there have been no changes to the previous references or reference order.

[Comments to PONE-D-23-13935 – revised]

1. Due to the small sample size of each age and gender subgroup, advanced analysis of age- and gender-specific mortality rates could not be applied. I could not find the “additional Fig 1 and 2” as indicated in the authors’ response. This limitation can be addressed in discussion. 

- Response: Thank you for the thoughtful suggestions of the reviewer. The additional Fig 1 and 2 are shown in the uploaded file "response to reviewers". 

We have incorporated this result into the limitation section of the manuscript. However, we did not present or cite the additional figures in the result section because this result lacks statistical power due to small sample sizes of each subgroups, so the result is difficult to explain and interpret. (Line 413-419)

2. To make it clear for readers, “age” could be changed to “age of onset/diagnosis”. 

- Response: Thank you for the detailed feedback that the authors missed. We replaced ‘age’ with ‘age at diagnosis (AAD)’ in the manuscript, and we cited an article (Ref: 26) that used the abbreviation ‘AAD’ for ‘age at diagnosis’. But, we did not replace ‘age’ in compound terms like ‘age-associated mortality’ with AAD. (Line 123-124) 

3. The authors have added subtitles to graphs, which makes the manuscript easier to read. Still, in the figure legends, all abbreviations should be illustrated. 

- Response: We thank you for pointing out the part the authors overlooked. We have fully illustrated the abbreviations which remained in the figure legends previously. 

4. Figure 3, what is “No. at risk”? Please use the original words instead of abbreviation, and indicate “age of onset/diagnosis) above the age group could be more illustrative.

- Response: We denoted the original term ‘Number at risk’ in Fig 3. Additionally, we added vertical columns named ‘age at diagnosis’ to the left of the number of AAD groups.

---

## [Editor Report · Decision Letter 2]

26 Oct 2023

Age-associated mortality is partially mediated by TERT promoter mutation status in differentiated thyroid carcinoma

PONE-D-23-13935R2

Dear Dr. Kim,

We’re pleased to inform you that your manuscript has been judged scientifically suitable for publication and will be formally accepted for publication once it meets all outstanding technical requirements.

Kind regards,

Avaniyapuram Kannan Murugan, M.Phil., Ph.D.

Academic Editor

PLOS ONE
---

## [Editor Report · Acceptance letter]

2 Nov 2023

PONE-D-23-13935R2 

Age-associated mortality is partially mediated by *TERT* promoter mutation status in differentiated thyroid carcinoma 

Dear Dr. Kim:

I'm pleased to inform you that your manuscript has been deemed suitable for publication in PLOS ONE. Congratulations! Your manuscript is now with our production department. 

Kind regards, 

on behalf of

Dr. Avaniyapuram Kannan Murugan 

Academic Editor

PLOS ONE